# A Case Study of Existing Peer-to-Peer Energy Trading Platforms: Calling for Integrated Platform Features

**Shan Shan** [1], **Siliang Yang** [2,*], **Victor Becerra** [3], **Jiamei Deng** [2] and **Honglei Li** [4]

1 School of Strategy and Leadership, Coventry University, Coventry CV1 5FB, UK; shan.shan@coventry.ac.uk
2 School of Built Environment, Engineering and Computing, Leeds Beckett University, Leeds LS2 8AG, UK; j.deng@leedsbeckett.ac.uk
3 School of Energy and Electronic Engineering, University of Portsmouth, Portsmouth PO1 3DJ, UK; victor.becerra@port.ac.uk
4 Department of Computer and Information Sciences, Northumbria University, Newcastle upon Tyne NE1 8ST, UK; honglei.li@northumbria.ac.uk
* Correspondence: s.yang@leedsbeckett.ac.uk

**Abstract:** The emergence of distributed energy has led to a change in the role of the consumer in the traditional sense over the past decade. The proliferation of emerging generators and distributors has created opportunities for a more decentralised and open energy market. In particular, the emergence of peer-to-peer (P2P) energy trading models, challenged by the surge in demand for sustainable energy, has eliminated the need for intermediaries in energy transactions between consumers, producers, and sellers. Due to the great promise of sustainable energy, both in terms of its contribution to the environment and production costs, this paper reviews a number of well-known P2P energy trading platforms to understand what makes P2P energy trading platforms more functional. As a result, areas for consideration were identified and grouped into five themes: (1) set-up, (2) market, (3) information, (4) price, and (5) regulation.

**Keywords:** decentralised energy market; P2P energy trading; functionalities

## 1. Introduction

The innovation in photovoltaics and wind turbines, allowing people to produce their own electricity, has made energy consumers become prosumers. Zhou et al. [1] explain this by recognising that operators allow for a local market platform with the facilities required for consumers to reap individual rewards by allowing prosumers to share and trade energy among themselves. This process has resulted in a new power system method with energy being traded on microgrids as people self-sufficiently supply their own electricity for homes, offices, and small industrial settings with renewable energy sources (RESs), which can be traded and shared in local areas [2–4]. This can form the basis for creating peer-to-peer (P2P) energy trading initiatives [5], involving prosumers selling excess energy generated by their own renewable energy sources to other users in the community. Up to date, P2P trading platforms have emerged in a range of sectors, allowing small suppliers to compete with traditional providers of goods and services [6].

A supply–demand equilibrium can be achieved through the share of energy, with prosumers with surplus energy supplying those with energy demand [7]. Earlier studies emphasise different dimensions in discussing P2P energy trading. In terms of the research focus, cost saving, efficiency, and community are the most discussed topics [8–10]. In terms of the research approaches, game theory [11,12], constrained optimisation [13,14], simulation [15,16], and blockchain [17,18] are the most used. The primary aim of this paper is to review the operational mechanisms underpinning P2P energy trading platforms and to deduce a set of integrated platform features that bolster functionality and sustainability. To achieve this, this research proposed the following specific research questions:

(1) What are the essential components that constitute an effective P2P energy trading platform, and how do they interrelate?

(2) How do existing P2P energy trading platforms integrate these components to meet the challenges of sustainable energy demands?

(3) What operational models can be abstracted from current P2P platforms, and how might they inform the development of future trading systems?

Thus, this research reviews projects on P2P energy trading as a potential mechanism that can play an important role in future electricity markets. Building on the current literature, we seek to review the fundamental components of P2P from stem to stern of an integrated operation. For instance, there are five primary components that a comprehensive P2P energy trading operation should encompass: the microgrid set-up, market mechanism, price mechanism, information mechanism, and legal/regulatory frameworks. Secondly, this study compares different energy trials within the components of the platform itself. For example, we consider the P2P energy trading trials from the year 2015 and select the most discussed projects to be the cases.

This study conducts an in-depth summation of recent research and practical applications in the realm of P2P energy trading, delineating its strengths, weaknesses, challenges, and potential future developments. Central to this research are three principal innovations and significant contributions. Firstly, it undertakes a comprehensive analysis of P2P energy trading platforms, an area previously examined primarily within narrow technological or economic confines. This expansive approach integrates a variety of dimensions—technical, economic, social, and regulatory, thereby fostering a more holistic understanding of P2P energy systems.

Secondly, the study meticulously identifies and explicates five pivotal components of P2P energy trading platforms. This elucidation provides a structured analytical framework, instrumental for evaluating and forging efficient and sustainable energy trading systems on a global scale. Furthermore, the research underscores the policy ramifications of P2P energy trading. It accentuates the necessity for flexible regulatory frameworks that can nurture the growth of renewable energy initiatives and adapt to evolving market needs.

Moreover, the study maps out prospective avenues for future research, highlighting critical areas such as the integration of emerging technologies, evolving consumer behaviors, and the dynamics of energy markets. The essence of this work lies in its thorough scrutiny of the operational processes of P2P energy trading, comparing and contrasting various schemes to illuminate the diverse mechanisms at play. This exploration not only enriches existing knowledge but also aids in discerning potential impediments that could impede the advancement of P2P trading schemes. Overall, this study makes a substantial contribution to the field of P2P energy trading, enhancing understanding and paving the way for future innovations and policy development.

## 2. Literature Review

### 2.1. P2P Energy Trading: A Step toward an Integrative Definition

Electricity is an essential component of daily life and has become a dominant feature of the power industry over the last century. Traditionally, electricity production involves the transmission of electricity from large-scale power plants through high-voltage lines into transmission grids for distribution to major clients, such as large businesses and industries. Subsequently, the electricity is converted to medium or low voltage before being distributed to local grids for smaller customers. All customer contributions are then incorporated into the supplier's bill, which is distributed according to the appropriate amounts.

The emergence of distributed energy resources (DERs) has challenged the entire power sector, as Calvillo et al. [19] explain. The DERs comprise operators and prosumers, with operators acting as coordinators for DERs, dictating centralisation or distribution in relation to the best economic advantage and providing ancillary power system services, such as frequency response and voltage support. Direct control is a key component of centralised coordination and is utilised to control demand-side resources like commercial

and residential thermostatically controlled loads (TCLs), energy storage systems, and electric vehicles. Prosumers are components that produce and consume electricity [1]. The major advantage of DERs is the ability for prosumers to dictate their individual DERs, without requiring motivational incentives. The proliferation of innovative DERs, such as wind turbines and PV cells, has led to an increase in the number of prosumers among energy consumers. This allows for the foundation on which to create P2P energy trading initiatives [5], meaning that prosumers who create more energy from their renewable energy sources than they can consume themselves are able to sell that excess to others within their local area. Sharing energy creates a better balance of supply and demand, with the prosumers providing the supply to those with the local demand for it. The site of renewable energy sources, such as wind turbines and solar panels, as well as weather conditions, can greatly influence their effectiveness, while if a significant amount of energy is introduced to the main grid, it runs the risk of causing a breakdown. However, the inclusion of small suppliers who can trade in energy can greatly reduce that risk, and the advances in battery and photovoltaic technology make that possible.

P2P energy trading can be properly defined as prosumers generating their own energy from RSEs in offices, factories, and homes, then sharing and selling it locally [2]. From this description, we can deduce that the term "prosumers", an amalgamation of "producers" and "consumers", comes from the fact that producers not only consume the electricity they use themselves but also sell their excess to others. With the development of DERs and the Internet of Things (IoT), the transaction was bilateral. As renewable energy and energy storage technologies have advanced, the balance of supply and demand is more easily managed and more clearly controlled, with modern DERs being predictable and allowing the trading between prosumers and consumers to work smoother and quicker.

P2P energy trading is therefore considered important for two main reasons. The first is the fact that it has enabled consumers to develop into prosumers with the help of DER technology. The production and supply of electricity continue to evolve and grow due to forward-thinking customers making moves to change and replace the traditional passive consumer market, which relies on traditional methods of energy supply. Most countries' energy supplies were controlled by a few very big companies that had a monopoly on electricity production and distribution, meaning consumers had no involvement or influence on the supply. However, the emergence of smart grid and information and communication technology (ICT)-based technology means that massive amounts of data can be gathered from various sources for analysis. These sources, which include IoT and wearable devices, as well as sensor networks [20], allow for bilateral communication between consumers and prosumers. This area has been examined by several researchers with the help of methods and devices such as smart meters, advanced metering infrastructure, home automation, and home area networks [21]. This technology not only makes energy more efficient, but the fact that it allows consumer data to be communicated to the suppliers means that individual electricity usage can be better assessed, coordinated, implemented, and adapted. Parag and Sovacool [22] explain that the use of smart meters and vehicle-to-grid electric cars allows consumers to assess their own energy usage and adapt this accordingly, whether through direct contact with suppliers or through trade with smaller agents more accurately.

P2P energy trading is still a relatively new and uncommon concept among regular consumers. In the late 20th century, Europe fully embraced ICT into its distribution power network, with the aim that all customers would eventually be able to trade the electricity generated through distributed power sources [23]. In October 2015, a UK-based company, Open Utility, launched a program whereby commercial energy users and renewable energy producers can trade directly through their energy trading platform and cut out the need to go through power utilities. Since its launch, services with similar models to Open Utilities have been fostered in other countries, like Transactive Grids in America and the German version Peer Energy Cloud [23].

### 2.2. Research Objectives and Approaches to P2P Energy Trading Research

To further substantiate the identified themes, this research incorporated empirical data, including case studies that illustrate the practical application and outcomes of these themes in real-world P2P energy trading platforms. During the first half of the 2000s, studies focused more on the concept and business model concerning P2P as well as examining certain legal concerns [24–26]. As the P2P networks developed during the latter half of the 2000s, studies focused more on P2P applications, including P2P energy trading. This was also when studies considered platform setting up as well as information systems, including the way in which the wireless network can use sensors to decrease energy consumption [27,28] and the differences between the traditional energy model and the new P2P architecture [29,30]. What is the architecture for the P2P energy trading system, and how does a distributed brokering system support a hybrid environment, including integrating the evolving ideas of computational grids, distributed objects, web services, P2P networks, and message-oriented middleware [31]?

Between 2010 and 2015, an increasing number of studies focused on P2P energy trading management and price mechanisms. For trading management, a study by Wu et al. [32] assessed a hybrid energy trading market involving an external utility company and a local trading market that a local trading centre managed. For the price mechanism, studies focused on minimising the overall energy cost and the P2P energy sharing losses in a distribution network consisting of multiple microgrids and explicitly incorporating the practical constraints (for example, power balance and battery operational constraints) to handle the mismatch problem between local demand and local generation in microgrids [33,34].

After 2015, P2P energy trading projects began to be developed, and case studies have sprung out in recent years [2,35], which are more about different objectives and research approaches. For the different objectives, research is mainly focused on cost savings, community building, and effectiveness. For cost savings, optimal algorithms were developed to minimise energy costs in P2P energy trading compared to traditional trading methods [8]. Energy trading directly between various DERs allows buyers to save costs and sellers to make a profit, thereby resulting in a win-win situation [36]. The effectiveness factor includes transaction effectiveness and energy effectiveness. Studies have highlighted the demand side management (DSM) system, which is designed in a way that users are encouraged to alter their usual energy consumption patterns when there are energy price changes in coordination with enhanced energy effectiveness in the smart grid DERs [37]. For example, in the microgrid, P2P energy trading can lead to the participating households experiencing unfair cost distribution. To resolve this, Pareto optimality [38] should be effectively used to optimise multiple factors so that no households suffer for improving another's cost [9]. Further, studies concerning transaction effectiveness tend to focus on the smart contract of blockchain. Smart contracts carry out the trading and payment rules strictly with no human interaction, and thus, there is greater security and fairness in energy trading. According to case studies regarding the Ethereum private chain, the proposed mechanism has evident benefits in terms of reflecting market quotations, facilitating the use of renewables, and balancing the profits of players. This encourages players to participate in the P2P energy trading. The authentic gas consumption, as well as computational time to the smart contract suggests that this platform can ensure effective and efficient transaction with multi-player participation [39]. Some studies focus on community cohesion and belonging, as the trading of energy within the community can add to community cohesion and the search for a sense of community belonging. Participants, for example, stated the ways in which the local community uses cooperative schemes for engaging people, building community cohesion, increasing community awareness, and helping groups to have decision-making power, responsibility, and ownership over the energy problems impacting them. Further, they noted how such schemes can be advantageous to local communities by providing investment opportunities, retaining money in the local economy, and ensuring steady income streams [40].

In terms of research approaches, five major approaches were identified: algorithms, game theory, simulation, blockchain, and constrained optimisation. Game-theory-based methods were used to analyse the competition and cooperation between DERs, resulting in stable and mutually beneficial solutions for different shareholders [10,16,41]. Simulation and algorithm optimisation techniques were also used to enhance the effectiveness of P2P energy trading [13]. The blockchain technology helped obtain a replicable data structure that could be shared among members, allowing transparent, decentralised, and secured P2P network energy trading [42].

Secondly, the five major approaches concerning the topic are algorithms, game theory, simulation, blockchain, and constrained optimisation. The game theory is used to capture the competition and cooperation between DERs to deliver a solution that is stable, sometimes optimal, and mutually beneficial for different shareholders. Some of the well-known methods are the coalition formation game [16] and the Stackelberg game [10,41] which help prosumers compare the advantages of P2P trading including and excluding its battery, thereby enabling prosumers to develop appropriate social coalition groups with similar prosumers within the network to carry out P2P trading [43]. Auction theory is also founded on the game theory, which involves known methods including double auction [17] and pool-structured auction that add a parallel and short-term pool-structured auction to the market layer features cleared by a new decentralised ant-colony optimisation method [12].

Simulations were converged, and game equilibrium was optimised through algorithms. Algorithms are also used by trading platforms, especially blockchain, for optimising processes when conducting energy transactions including the near-optimal algorithm and energy cost optimisation via trade that coordinates P2P energy trading in smart homes that have a DSM system [13].

In terms of mathematical programming techniques, constrained optimisation is utilised to enhance the P2P energy trading parameters under the market and power system's various hard and soft constraints. For example, in a study by Lüth et al. [44], a linear programming (LP) approach was used for designing an innovative multi-energy management strategy following the multi-energy demand's complementarity for investigating prosumers' optimal energy scheduling problems.

Blockchain helps obtain a data structure that is replicable and that can be shared with members, thus allowing transparent, decentralised, and secured P2P network energy trading. In the study of Schneiders and Shipworth [42], for example, the UK was considered as an example to determine if UK-based energy communities are able to address the uncertainties concerning individual energy consumers using blockchain for P2P energy trading.

The latest research explores various components of P2P energy systems, such as market mechanisms, regulatory frameworks, and information mechanisms [45]. A thorough examination of the current status of P2P energy sharing is presented in another comprehensive review, which discusses the advantages, obstacles, and different technological and business models that have been created to support P2P energy trading [46]. Simultaneously, a comprehensive evaluation of the current progress and outlooks in the domain of P2P electricity trading was conducted. Through comparative analysis of diverse P2P energy trading schemes, scrutiny of their business models, market focus, and utilisation of blockchain technology, the potential of P2P energy trading to offer enhanced efficiency and sustainability in energy systems was established [4]. However, these studies are still limited on different research objectives and approaches to achieving P2P energy trading. There are rare papers that review the literature from the platform itself, including different components, while bottom up from the start of the set-up of the platform to the end regulation discussion is another dimension of reviewing the topic.

## 3. Case Studies

The criteria for the inclusion and exclusion of platforms were based on a broad spectrum of operational models, geographic diversity, and technological variations. This

research specifically targeted platforms that have been actively engaged in P2P energy transactions for a minimum of one fiscal quarter, thereby ensuring the inclusion of only those platforms with demonstrable activity and excluding nascent or speculative initiatives. Additionally, we sought platforms that have undergone regulatory scrutiny or peer-reviewed validation to ensure the integrity of our analysis. Conversely, the exclusion criteria were equally stringent, disallowing platforms that lacked transparent operational data or those that functioned solely within the confines of a microgrid without interfacing with broader market mechanisms. This bifurcation enabled a focus on platforms that offer insights into the scalability and interoperability of P2P trading systems.

In the pursuit of a comprehensive analysis of P2P energy trading platforms, this study utilised a diverse array of data sources, ensuring a rich and multifaceted understanding of the subject. Primary among these were academic journals and conference proceedings, which provided peer-reviewed research findings and theoretical perspectives. These academic sources were complemented by industry reports and white papers from leading energy firms and regulatory bodies, offering insights into practical applications, market trends, and regulatory frameworks. Furthermore, case studies of existing P2P energy trading platforms formed a cornerstone of our research. These case studies were meticulously selected to represent a broad spectrum of operational models, technological infrastructures, and geographic contexts.

The case methodology is microscopic due to the limited number of cases. In fact, the relative size of the sample, for example, whether 2, 10, or 100 cases are used, does not transform a multiple-case study into a macroscopic study to meet the established objectives [47]. Here, in this research, we reviewed and compared these different cases under the same P2P energy trading operation framework of a macroscopic study.

There are various P2P projects in the market. According to Google Trends, the distribution countries of interest are mainly the United States of America, Canada, European countries, Australia, South Africa, and India. Thus, the review projects were selected from these highest-interest countries. Also, in terms of the period selection, the interest trend of P2P energy trading started from 2015 to now, as shown in Figure 1. This study selected the projects formally launched since 2015. Thus, the case study ranged from 2015 to now and included eight trading trails, namely, Piclo in the UK, Vandebron and Vattenfall Powerpeers in the Netherlands, SunContract in Slovenia, Power Ledger in Australia, Brooklyn Microgrid in the USA, Noncommunity in Germany, and Brazilian Energy Communities in Brazil.

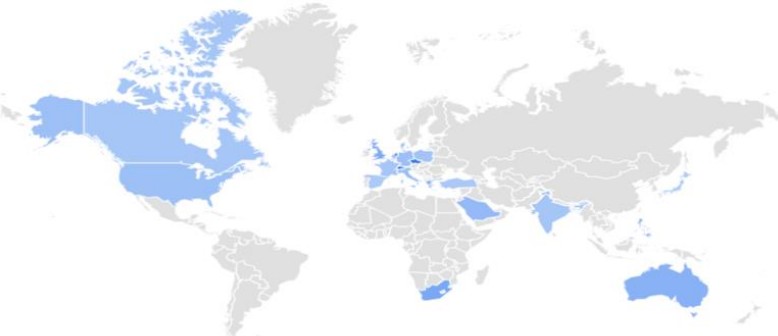

**Figure 1.** The distribution of interest in P2P energy trading topic over time (available from Google Trends). The blue colour represents projects covered areas, and the grey colour represent projects uncovered areas.

In the second phase, the study drew on cross-sectional in-depth case studies on the use of P2P energy trading on the different P2P energy trading trails.

### 3.1. Case Study: Piclo

Piclo was founded in response to the increasing energy consumption from commercial consumers and the proliferation of independent suppliers of renewable energy. Piclo is the first P2P market platform for renewable energy in the UK. Both suppliers and consumers can subscribe to Piclo free of charge. The platform employs advanced algorithms to match user demand to a close-by generator based on preferences and locations, as well as offering customers data visualisations and analytics at intervals of half an hour. The energy supplier Good Energy is responsible for the provision of contracts, data metering, billing, customer service, and marketplace information. Such data are helpful in customer interaction, proving sustainability or provision of rewards for the purchase of renewable energy via Piclo Flex—an independent marketplace [48,49].

As shown in Figure 2, a large hydrogenerator initially supplied the greatest proportion of energy [48], but advances in technology and the growing proliferation of DERs (for example, solar panels, batteries, electric vehicles) have subsequently afforded sellers the possibility to combine different types of energy. Limits on allocation and prices can be imposed by producers, while the costs associated with the installation and maintenance of local distribution networks can be recovered via the distribution use of system (DUoS). The amount of electricity going through the distribution network and being used determines the charges, with variation in rates according to location and time of day [50]. In a white paper published by Open Utility in 2018, Piclo presented the DUoS model, in which the discounted rates were applied for demand–supply matches, offering a much weaker price signal compared to the Network Replicating Private Wires (NRPW) to promote matching at a local level. The reason for this is the necessary amendments to regulations, legislation, and industry code. A strong signal for matching at a local level can be provided by the NRPW, but the model is associated with notable obstacles to entry, such as upfront capital costs, the complex nature of contracts, and a wide range of logistical difficulties. Therefore, this model is not suitable for a large number of UK users. Furthermore, scheme implementation can be problematic due to regulation as well, because retailers are required to obtain a licence and comply with the licensing agreement. Suppliers with over 250,000 accounts have to make a contribution to the Energy Company Obligation and manage the feed-in-tariff payments of customers with solar panel installations [51].

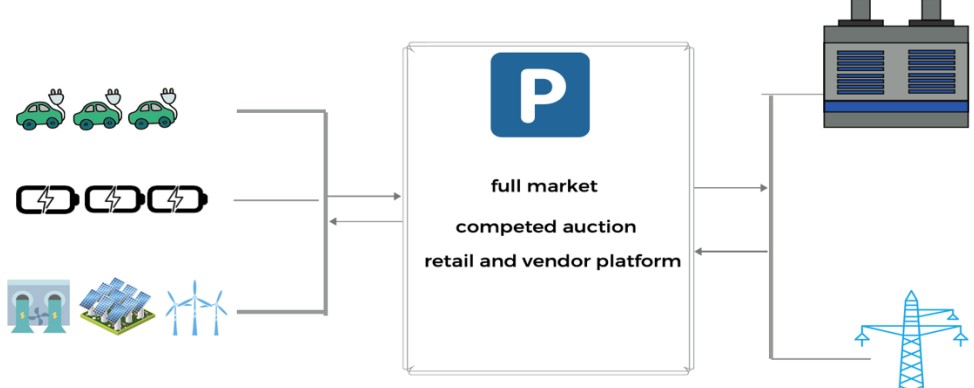

**Figure 2.** Operation model of Piclo.

### 3.2. Case Study: Vandebron

Vandebron was established in the Netherlands in 2014. It is an energy company that purchases renewable energy from local solar farms and large wind parks and distributes electric energy and gas to independent suppliers of renewable energy and consumers at a flat rate of USD 12 [52]. In 2017, Vandebron initiated a collaboration with Alfen, a supplier of smart electric vehicle charging stations for home and business, with the purpose of achieving energy grid equilibrium with blockchain technology [53].

With regard to pricing, energy producers are free to establish their own prices, while the customers can select which producer to use (Figure 3). Vandebron is also a supplier of $CO_2$-compensated natural gas. Customers can select from six compensation projects in different developing countries that they desire to support with their gas consumption [54]. Based on this P2P platform, producers obtain higher energy rates, whilst consumers know that their money goes towards an increase in the production of local and renewable energy. In February 2016, the Vandebron website listed around 50 energy producers that were capable of meeting the energy needs of over 30,000 households [22]. In December 2018, the number of households supplied by wind, biomass, and solar energy reached 100,000 [55].

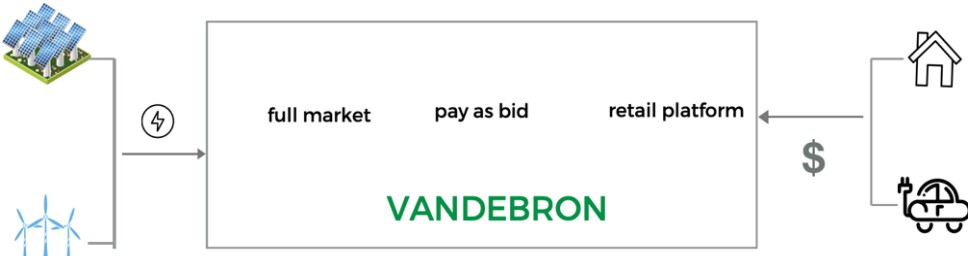

**Figure 3.** Operation model of Vandebron.

### 3.3. Case Study: SunContract

SunContract was established in Slovenia on 13 April 2018. It is a live blockchain-powered P2P platform operating nationwide and promoting the free purchase, sale, or trade of electric energy among individuals. As shown in Figure 4, the overall goal is to achieve better energy autonomy and sustainability through the development of a self-sufficient and renewable energy community with minimal environmental impact [56]. To this end, the company has created an energy pool that encompasses both producers and consumers of electricity to encourage trading. The pool enables participants to select energy sellers and buyers according to their own preferences.

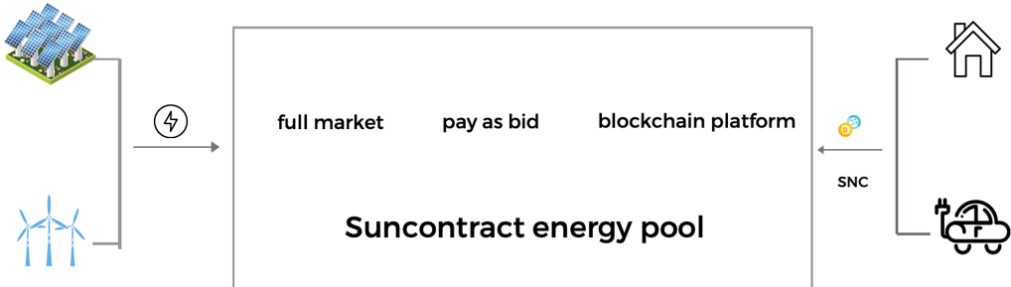

**Figure 4.** Operation model of Suncontract.

Aside from energy suppliers, SunContract customers can also choose the price they are financially and environmentally willing to pay. The customers' personal dashboard allows them to track how much energy they use per day and make adjustments accordingly. Likewise, SunContract permits energy producers to set the sale price for their excess energy as well as double auction bids on the marketplace for consumer orders [57]. For the purposes of electric energy trading, SunContract tokens (for example, Ethereum-based tokens) are necessary, being sold as software and energy licence that the registered customers can then use in the energy pool. Furthermore, to ensure that the cost of electric energy is covered for a minimum of two months, all the platform users must have 1000 or more such tokens at all times [58]. Thus, SunContract aims to link autonomous supply and demand via the energy pool underpinned by smart contracts by substituting the current middleman role with blockchain technology. In this way, the transparency of the producer–consumer interactions can be guaranteed through the pool [56].

### 3.4. Case Study: Vattenfall Powerpeers

Vattenfall founded the social media energy platform Powerpeers in the Netherlands in 2016. This is the first digital and interactive marketplace in Europe, aiming to integrate supply and demand for self-generated energy at a local level [59]. Between 2016 and 2018, the platform saw a considerable increase in contracts, from 16,000 to over 60,000 [60]. The scheme seeks to make the share of energy as straightforward as making an Airbnb booking and affords customers options regarding producers and consumers of self-generated energy [61]. As shown in Figure 5, the Powerpeers platform facilitates the selling and buying of energy, with customers being allowed to choose their personal supply mix from up to ten distinct sources, which are completely transparent, and they can also invite friends to participate in energy sharing. Producers can be selected according to location, type of energy, or description, and there are no restrictions on producer choice, provided that supply is still available. Moreover, customers can change their mind about their preferred producers every day [51].

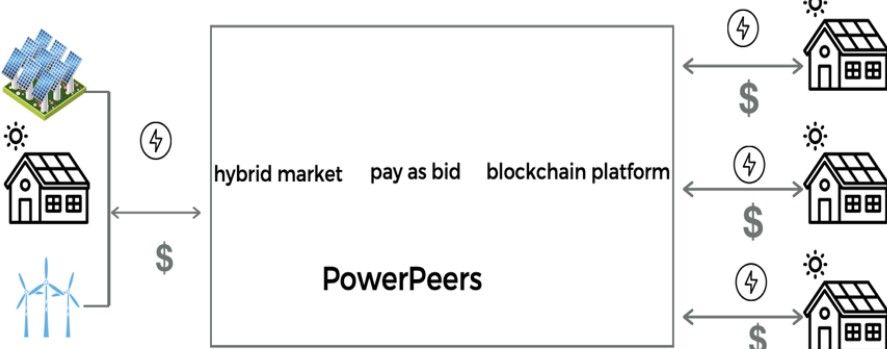

**Figure 5.** Operation model of Powerpeers.

As a customer-based platform, Powerpeers is underpinned by the many-to-many model, whereby consumers can have more than one energy supplier, while energy suppliers can have more than one consumer. Hence, Powerpeers introduces internet interaction within the energy market. This is what differentiates Powerpeers from traditional energy suppliers, who employ a one-to-many model [61]. Essentially, the platform undertakes the digitisation of each kWh and offers almost real-time matching of supply and demand. Furthermore, the platform eliminates the necessity for energy certificates as it can precisely pinpoint the source and amount of energy [51].

Essentially, subscribers can both produce and use energy. For a monthly fee, they can sell their self-generated energy to others as well as select their preferred sources of energy [59]. Price signals can be issued in the community since individual households can sell their self-generated renewable energy at their desired price, as well as the fact that they pay for network connection and use. In this regard, no regulatory obstacles should arise, which makes the platform capable of the required development [51]. Nevertheless, it must be highlighted that, despite its licensing availability, Powerpeers is a private platform and may not be suitable for nationwide or worldwide implementation in the future, according to proponents of public or open platforms [51].

### 3.5. Case Study: Power Ledger

Power Ledger was founded in Australia in 2016. It is a P2P energy trading scheme that employs blockchain technology to facilitate market trading and clearing mechanisms [62]. As shown in Figure 6, it is intended to enable residential and commercial units in the grid or act as microgrids on their own to share their excess energy [63] at their desired price. Energy trading across the distribution network requires giving a portion of profits to distribution system operators [64]. Furthermore, Power Ledge allows battery sale, storage,

and use. For instance, the SkyHomes system uses an embedded electricity network, solar PV, and storage microgrid to supply residential units with fully renewable energy [65].

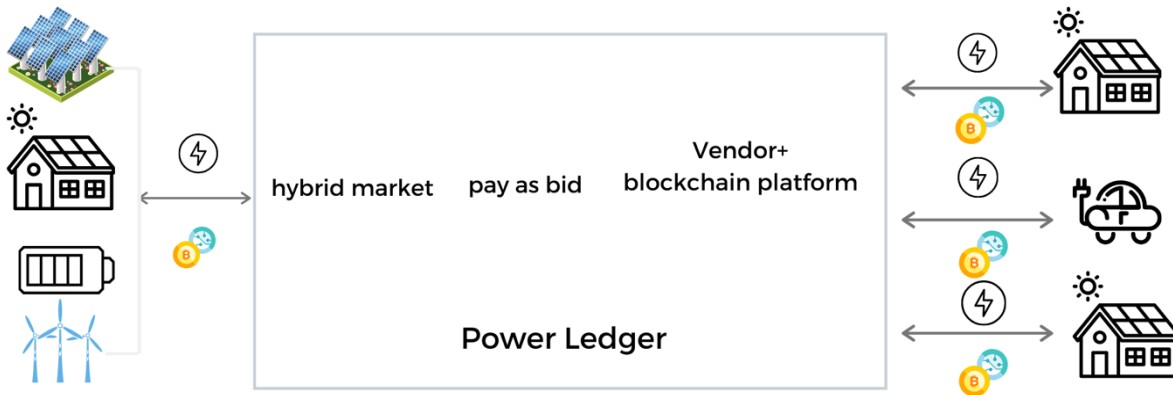

**Figure 6.** Operation model of Power Ledger.

Power Ledger, a company focused on increasing the availability of renewable energy for residential units, has developed an asset ownership model that allows for greater activation of renewable energy assets [63]. This model is underpinned by a P2P energy exchange platform, which is built on the Ethereum network contracts. The primary aim of this platform is to democratise energy trading and enhance system transparency to enable individuals and communities to make optimal decisions regarding their energy supplier. To further promote a fair environment for exchanging energy between self-generating residential units and neighbours, Power Ledger has introduced a dual token model. This model requires users to escrow power as a utility token to participate in the platform, while Sparkz, which represent electricity credits associated with local fiat, are used by marketplace participants [66].

### 3.6. Case Study: The Brooklyn Microgrid

The Ethereum-based photovoltaic energy trading platform called the Brooklyn Microgrid emerged as a demonstration project in April 2017, in Brooklyn, New York [35]. The project involved 50 Brooklyn residents, who installed solar panels and sold excess energy. The Brooklyn Microgrid case can be illustrated in Figure 7. The platform allows prosumers to decide whether to engage in P2P trading of their self-generated solar energy, store the excess energy in an online or offline storage device, or use the energy for household purposes. In other words, prosumers with solar panel installations generate electric energy and can sell their excess energy to other consumers. The energy transactions among neighbours are monitored by a blockchain network, with household smart meters indicating activities of energy production and use [66]. The grid is linked to the broader commercial grid and employs the TransActive Grid platform to ensure the security of customers' purchase and sale interactions, which can occur automatically based on smart meters and smart contracts [67]. Therefore, the Brooklyn Microgrid achieves the integration of the online energy market community and the physical microgrids. During the initial phases of the project, the TransActive smart meters were confirmed by the analogous meter. Moreover, to avoid power disruptions, physical microgrids were established alongside the existing grids [35].

Active participation in the local energy market is possible for a restricted number of prosumers and consumers, with documentation and verification of energy transactions between them being achieved through a private blockchain-based distributed ledger technology network [68]. Furthermore, a double-auction mechanism is adopted by the Brooklyn Microgrid, so that essential infrastructure (for example, hospitals) can obtain energy at fixed rates, while residential and commercial units are required to bid on the rest of the available energy [35].

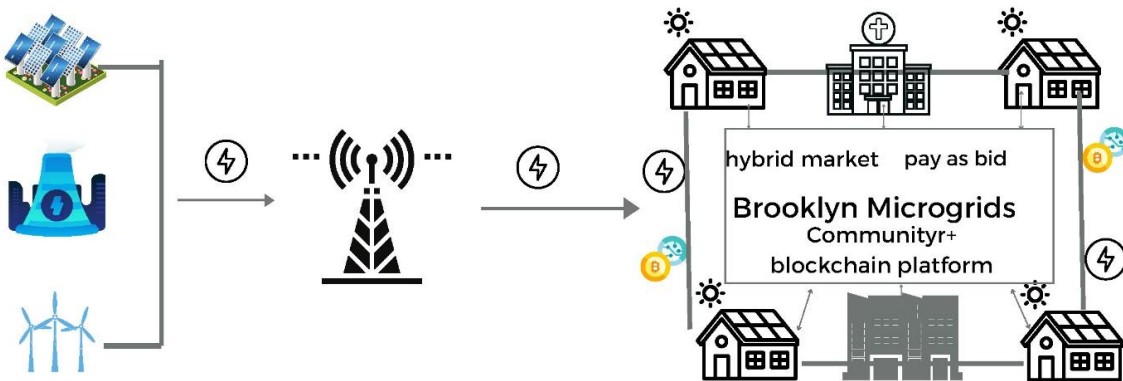

**Figure 7.** Operation model of Brooklyn Microgrid.

The Brooklyn Microgrid utilises a private blockchain with the Tendermint protocol to offer various business models for distributed grid and transactive energy space [64]. Revenue sources for the Brooklyn Microgrid include the sale of energy storage units and commissions from energy trading [55]. However, the regulatory framework for local energy trading is governed by the legal milieu, and current regulations in the Brooklyn area do not encompass local P2P energy trading without utility participation. To address this, the Brooklyn Microgrid is collaborating with utilities and is poised to obtain a licence as an energy retailer in the near future [35].

### 3.7. Case Study: sonnenCommunity

The sonnenCommunity was founded in Germany in 2016. It is a platform for renewable energy trading. The sonnenCommunity is geared towards using a central software connecting and keeping track of all community members to achieve an equilibrium between the supply and demand of energy [69]. As shown in Figure 8, the set-up phase involves network integration of various types of energy, such as solar, wind, and biomass energy. For power grid stabilisation, distributed network operators ensure instant delivery and supply of energy. Thousands of sonnenBatterie (the smart control centre of the sonnenCommunity) users are connected by a sonnen-Flat-Box, with enormous battery pools being created through the integration of a large number of separate home storage units. The "Sonnen-Flat-Box" refers to the SonnenCommunity's energy storage and management system, which allows homeowners to store and manage their own solar energy, as well as share excess energy with others in a peer-to-peer network. The massive volumes of energy accumulated through such "virtual storage" contribute to the stabilisation of the public power grid [70].

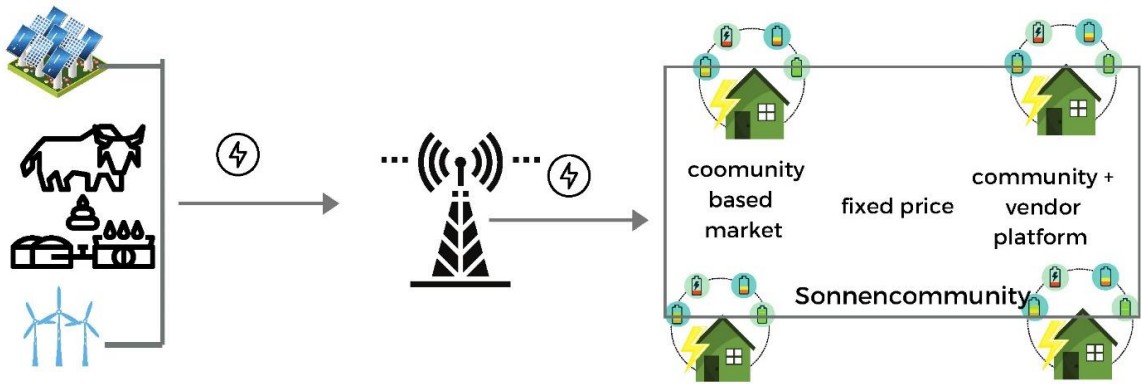

**Figure 8.** Operation model of the sonnenCommunity.

The sonnenCommunity sets a fixed price and offers a constant supply of energy based on battery storage technology. Prosumers earn 23 cents for every kWh, and platform

subscription costs EUR 19.99 monthly, without any transaction fees or other extra costs [55]. Furthermore, during days with high sunlight levels, consumers can fully meet their energy needs on their own based on the sonnenBatterie and photovoltaic system. Any excess energy produced is stored in a virtual energy pool that other members can use when unfavourable climatic conditions interfere with energy production, rather than being transferred to the main power grid [69]. In other words, by integrating energy trading and battery storage technology, the sonnenCommunity enables the creation of excess energy that can be stored in batteries for emergency situations or to be sold at a subsequent time.

As a smart control centre, the sonnenBatterie facilitates optimisation of solar energy consumption. For instance, in the morning, when energy use is high, but production is low, the sonnenBatterie enables the use of the energy stored on the day before, whilst in the afternoon, when energy production is maximal, the sonnenBatterie stores the greatest proportion of the produced energy [69]. In Germany, licensing is not necessary. However, in other countries, there are various regulatory frameworks that prevent the installation of sonnenBatterie in residential areas. Thus, whereas numerous residential units in Germany are equipped with sonnenBatterie, this may not be possible in another country, for example, Australia, where the laws regarding installation are extremely stringent [69].

### 3.8. Case Study: Brazilian Energy Communities

In Brazil and across Latin America, energy communities have historically developed in three distinct scenarios, as elucidated by [71,72]. Firstly, rural areas lacking public utility services often see consumers, typically through rural cooperatives, establishing their own distribution networks or power generation facilities. Secondly, isolated communities, geographically remote from public utilities, necessitate the creation of self-sufficient microgrids and energy sources. Thirdly, prosumers with access to public utilities participate in electricity trading via net metering programs. Historically, the first two types of communities have played a crucial role in providing electricity access to segments of the population not served by conventional power providers. The Brazilian Energy Communities (Figure 9), as detailed in the work of [73], represent a novel approach to renewable energy trading within the framework of microgrids. Established with a focus on P2P energy exchange, these communities are designed to harness and distribute renewable energy sources, primarily solar power, across Brazil. This integrated approach involves a sophisticated set-up phase where diverse renewable energy sources are interconnected [74].

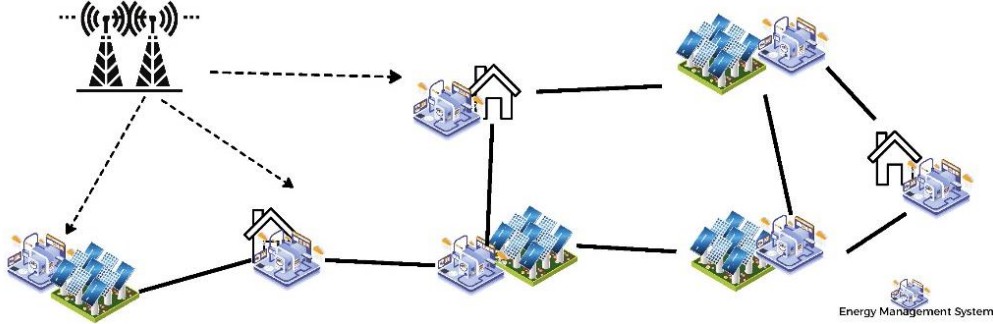

**Figure 9.** Operation model of the Brazilian energy communities.

The primary goal is to achieve a sustainable and self-sufficient energy ecosystem and circular economy within these microgrids [22]. This system not only ensures a consistent supply of renewable energy but also contributes significantly to the stability of the wider power grid in Brazil. The Brazilian model is characterised by its communal energy pools, where excess energy generated by individual members, especially during peak production periods, is stored for communal use [73].

In terms of pricing, the Brazilian Energy Communities adopt a model that encourages participation and fair compensation to further cope with the circular economy based on

the local level [22]. Prosumers are incentivised through a structured pricing scheme that rewards them for their contributions to the energy pool by a blockchain account. This strategy ensures an equitable distribution of benefits among community members and promotes a more widespread adoption of renewable energy practices.

From a regulatory standpoint, the Brazilian Energy Communities operate within a framework that encourages the development of such P2P energy trading models. While regulatory challenges may vary across different countries, the Brazilian model demonstrates a significant potential for scalability and adaptation in diverse legislative environments. In the context of Brazil, the primary impetus for embracing renewable energy initiatives is predominantly economical in nature. Nonetheless, it is challenging to distinctly separate the economic incentives from environmental considerations, as the most viable and competitive options for small-scale generation predominantly hinge on renewable sources [75].

## 4. Key Components of P2P Energy Trading

The market mechanism component of P2P energy trading encompasses market allocation and payment regulations and requires a bidding format with clear terminology to ensure compatibility between purchase and sale orders, thus enabling efficient distribution of traded energy [35,76]. To liberalise electricity supply and demand, a retailing spot market for electric energy is recommended to coordinate the demands of stakeholders [77]. The P2P market is categorised into community-based, hybrid, and full markets based on the level of decentralisation and topology, with the full P2P market involving direct trading among DERs based on multi-bilateral economic dispatch [78]. However, differences in prosumers' priorities and considerations regarding environmental and economic costs can result in challenges in reaching a consensus on the final energy delivery [79,80]. In contrast, the community-based market relies on a community manager to oversee the trading activities and mediate between the community and the overall system, allowing for more resilient community-based collaborations [81]. However, managing expectations and data while ensuring equitable energy sharing poses a challenge [82,83]. The hybrid P2P market combines features of both full and community-based markets and aims to provide greater predictability for grid operators and system compatibility for the future [84].

The wholesale price of electricity, including uniform and pay-as-bid prices, is determined by the supply of the transmission system operator and consumption [85,86]. Prosumers can adjust their loads to periods with low net demand, based on time of use (ToU) prices [87]. However, the real value of demand flexibility is not fully realised in P2P energy trading, as the market's small-scale DERs can impact dynamic pricing frameworks [88]. The auction-based framework and bilateral contract-based framework are the two major dynamic pricing frameworks, and the Vickrey–Clarke–Groves auction model with Bayesian game theory has been proposed to improve utility for prosumers and avoid negative utility for bidders in auctions involving multiple sellers and buyers and 15% power loss [89]. Due to the diversity in DERs and prosumers' preferences, product-related price differentiation can affect alterations in the bilateral contract-based framework, allowing for a dynamic value allocation to energy content and extra electricity features.

The fourth dimension of P2P energy trading platforms pertains to the information mechanism, which establishes participant connections, hosts the market platform, and ensures market operation surveillance [90]. Based on the nature of their information mechanisms, energy trading platforms are categorised into four types: retail supplier platforms, community platforms, blockchain platforms, and vending platforms [91]. By purchasing electricity directly from the wholesale market, electricity users can benefit from competitive commodity prices and avoid expenses associated with promotion, customer service, and advertising [92]. In contrast, vendor platforms aim to help distributed energy resource (DER) vendors enhance the value of their products. Battery storage and photovoltaic (PV) panel innovations offered by specific battery systems and PV panel vendors can reduce the charging cost for electric vehicle fleets and the functional loss related to high-voltage electricity [91,93].

Community platforms largely focus on identifying the benefits of separating microgrids from the main grid. At the community level, energy projects can focus on a common resource or goals such as improving supply–demand coordination to ensure greater energy efficiency of renewable energy [94,95]. DER coordination is also considerably important for ensuring that supply is not hindered when separating microgrids from the main grid. Common goals such as improving pollution at a local level can encourage prosumers to organise community energy schemes including microgrids [96] while also highlighting the importance of P2P energy trading platforms.

Energy trading platforms that are developed as per blockchain technology are known as blockchain platforms. Studies have focused on different forms of blockchain implementation concerning energy trading, and there has been increased attention to subjects concerning security and privacy related to decentralised energy trading [14,97]. Blockchain is an innovative technology involving transparency and ease of use, thus achieving effortlessness in generating and consuming energy [98]. Trading speed and efficiency can also be improved by using credit-based payment systems that will eliminate the need for blockchain using a reliable intermediary in P2P energy trading [97].

Regulations are a significant aspect that must be addressed in the development of P2P energy trading systems. In order to ensure a level playing field for platform-based businesses, retailers, and traditional utilities, regulators need to establish reforms in the current energy policy and laws. The recent EU Directive 2018/2001 concerning the use of renewable energy defines P2P trading of renewable energy for the first time and stipulates that market participants can engage in the sale of renewable energy through a contract with predetermined conditions concerning automated transaction execution and settlement, directly between market participants or indirectly via a certified third-party market participant including an aggregator [99]. While blockchain platforms enhance system security by applying a time stamp that makes data manipulation challenging, there is still a lack of legal clarification on how the recent General Data Protection Regulation (GDPR) applies to blockchains, putting users' privacy at risk [42]. Urgent regulatory frameworks must be promulgated to address this issue. Although China has implemented a top-down approach to private energy trading schemes, P2P markets primarily focus on consumers and involve a bottom-up approach where consumers can choose electric energy sources themselves [84]. Europe has been making strides in this subject and has taken the lead in developing the required regulations to expand P2P trading. For example, in January 2019, ten research and development initiatives focused on P2P markets were launched in Europe, while only one was initiated in the US.

## 5. Discussion

The seven cases above explained the details through the perspective of the P2P energy trading operation model. As shown in Table 1, the results were demonstrated under the five components, namely the grid set-up, market mechanism, price mechanism, information mechanism, and regulations. Table 2 presents a detailed comparative analysis of the selected case studies, which are pivotal in elucidating the operational intricacies of P2P energy trading platforms. Each case study is evaluated against the five key components identified in our research: the platform set-up, market mechanism, price mechanism, information mechanism, and legal/regulatory frameworks. This evaluation provides insights into how these components are manifested in different real-world scenarios and their interplay in the functionality of P2P platforms.

For instance, by examining the "platform set-up", we observe variations in grid integration and energy distribution strategies across the platforms, reflecting diverse approaches to managing energy supply and demand. For example, different approaches towards trading access are adopted by the reviewed schemes. The approach used by Piclo and Vandebron involves the identification of suppliers and establishing a price match for prospective customers. By contrast, SunContract achieves supply–demand coordination based on its own energy pool. All schemes focus on electric energy supply, apart from

Vandebron, which supplies gas as well. On the downside, the review could not gain significant insight into customers' perception of emotional connectedness to a particular supplier or into the factors influencing customers in supplier selection. Buyers can obtain energy straight from autonomous producers (for example, farmers with wind turbines). Vandebron is similar to Piclo in that it plays the role of an energy supplier offering incentive tariffs to encourage the exchange of energy between consumers and producers. Furthermore, by giving their excess energy to Vandebron, prosumers can buy energy from the company at a more affordable cost than from other suppliers. Meanwhile, the P2P trading mechanism implemented by Power Ledger, sonnenCommunity, and Brooklyn Microgrid involved prosumers exchanging their excess energy with their neighbours. By contrast, the mechanism adopted by Vandebron, Piclo, and Powerpeers was based on energy matching, with prosumers being allowed to select local producers of renewable energy. Similar to the model adopted by the sonnenCommunity in Germany, the Brazilian Energy Communities utilise a centralised software system to manage and balance the energy supply and demand among their members. And also, the Brazilian Energy Communities often focus on rural and isolated regions, with an emphasis on self-sufficiency, using predominantly solar energy [73].

The "market mechanism" comparison reveals how different platforms navigate the complexities of energy trading, with some employing advanced bidding systems while others rely on more straightforward peer-to-peer transactions; for example, the complete market model permitting direct P2P trading was adopted by Piclo, Vandebron, SunContract, and Brooklyn Microgrid. In this context, the platforms prioritise price matching and assisting both commercial and residential consumers in identifying the most suitable deal and price. On the other hand, a hybrid market model, integrating features from the full market and the community market, is adopted by Power Ledger and Powerpeers. This can facilitate direct trading among DERs whilst also making energy use and demand at the community level more sustainable. While Brazilian Energy Communities tend to be community-driven, with a strong emphasis on local cooperation and energy sharing. This collective storage and distribution mechanism echoes the "virtual storage" concept of the sonnenCommunity, where energy surplus is utilised to maintain grid stability [100].

In terms of the "price mechanism", the table highlights the diverse pricing strategies used, ranging from dynamic pricing models that respond to real-time supply and demand to more static models. This diversity underscores the importance of flexible pricing structures in accommodating varying market conditions. Most schemes adopt the pay-as-bid price model, although the Brooklyn Microgrid scheme uses a fixed-rate model, with constant prices for essential infrastructure. Acting as energy suppliers, Piclo and Vandebron implement a price match model, whereby consumers and producers are given incentive tariffs to promote energy exchange. This enables consumers to buy energy from close-by producers at a more affordable cost compared to other energy suppliers. Meanwhile, Powerpeers and SunContract allow individuals to purchase solar power plants, heat pumps, and power storage units, as well as afford them a range of options regarding energy products and services. Different from the above, Brazilian Energy Communities have pricing models tailored to local economic conditions, potentially with subsidies or incentives for renewable energy [73]. On the other hand, the sonnenCommunity applies a fixed-rate price for battery-derived energy, and therefore, consumers have to assume some risk when investing in such energy. Nevertheless, sonnenCommunity assumes the wholesale price risk, so consumers are unaffected by that risk.

The analysis of the "information mechanism" sheds light on the role of technology in facilitating energy trading. Some platforms leverage blockchain technology for enhanced security and transparency, whereas others utilise conventional digital platforms for transaction management. Vandebron and Piclo operate as retail supplier platforms focusing on assisting suppliers to keep their customers, as prosumers can obtain greater value from their DERs. P2P platforms afford suppliers better understanding of their customers so that more appropriate producer contracts can be drawn up. Although it adopts an approach similar

to Piclo and Vandebron, the sonnenCommunity also puts emphasis on the significance of a system of storage [2]. Hence, sonnenCommunity can be construed as a retail supplier as well as a vendor platform. In terms of blockchain technology, it is adopted to different extents by the various schemes, from complete adoption in the case of sonnenCommunity and Powerpeers to partial adoption in the case of the Brooklyn Microgrid and Power Ledger, which offer additional trading services.

Lastly, the "legal/regulatory frameworks" column in Table 2 illustrates the varying degrees of regulatory compliance and adaptation required by these platforms, reflecting the diverse legal landscapes they operate within. The commercial operation of microgrid energy markets is crucially dependent on the adoption of relevant regulations. Most countries accord great importance to this matter, but there is no standardised regulatory framework related to renewable energy. In the UK, regulation is stringent to ensure data and privacy protection. By contrast, although regulation for data protection has been introduced in other countries, in Europe, there is no stipulation about the installation of renewable energy infrastructure (for example, PV panels and battery storage). Meanwhile, Brazilian Energy Communities operate within Brazil's specific regulatory context, which is more supportive of community-led initiatives and focused on rural electrification [75].

Thus, by providing a detailed examination of these components across different case studies, Table 2 not only validates our thematic framework but also offers a comprehensive understanding of the operational models in P2P energy trading. This analysis is instrumental in identifying best practices, potential challenges, and areas for future development in the field.

**Table 1.** Platform components in P2P energy trading.

| Dimension–Platform | Reference |
|---|---|
| Set-up | [22,30,91,101–103] |
| Market mechanism | [16,35,76,77] |
| Information mechanism | [42,93,95,97,104] |
| Price mechanism | [12,13,43,87] |
| Regulations | [42,84,99] |

**Table 2.** Comparative analysis of current P2P schemes regarding the operation process.

| Project | Country | Year | Grid Set-Up | Market Mechanism | Price Mechanism | Information System | Regulations |
|---|---|---|---|---|---|---|---|
| Piclo | UK | 2015 | - Commercial<br>- Electricity | Full market | Pay-as-bid-competed auction | Retail supplier platform + vendor platform | Private data protection |
| Vandebron | Netherlands | 2014 | - Commercial/residential<br>- Electricity/gas | Full market | Pay-as-bid price match | Retail supplier platform | Not specific |
| sonnenCommunity | Germany | 2016 | Electricity | Community-based market | Fixed rate | Vendor platform + community platform | No licensing requirement |
| SunContract | Slovenia | 2018 | - Residential<br>- Electricity | Full market | Pay as bid | Blockchain platform | Not specific |
| Brooklyn Microgrid | USA | 2017 | Electricity | Full market | Pay as bid/fixed rate (e.g., hospital) | Community platform + blockchain platform | Not specific |
| Power Ledger | Australia | 2016 | Electricity | Hybrid market | Pay as bid | Vendor platform + blockchain platform | Specific requirements for installation of power batteries |
| Powerpeers | Netherlands | 2016 | Electricity | Hybrid market | Pay-as-bid price match | Blockchain platform | Not specific |
| Brazilian Energy Communities | Brazil | 2021 | - Residential<br>- Electricity | Community-based market | Local-economy-driven price match | Blockchain platform | Not specific |

## 6. Conclusions

In concluding our exploration of P2P energy trading platforms, this research illuminates the intricate interplay and integral nature of five key components within these systems: the platform set-up, market mechanism, price mechanism, information mechanism, and legal/regulatory frameworks. The findings underscore the necessity of an integrated approach where these components work in concert, highlighting the importance of adaptability, transparency, and responsiveness in effective P2P energy trading platforms.

A significant insight from the comparative analysis is the diverse integration strategies across various platforms, reflecting an array of approaches to sustainable energy challenges. However, a commonality in successful models is their cohesive alignment of these components, fostering a sustainable and efficient trading environment. The operational models this research studied offer blueprints for forthcoming systems, showcasing decentralisation potential through microgrids and the growing significance of technologies like blockchain for enhanced security and transparency. Flexibility, adaptability, and technological innovation emerge as key themes, pivotal in shaping future P2P energy trading platforms to meet evolving market demands.

## 7. Limitations and Prospects for Further Studies

This study's findings, while illuminating, are inherently delimited by the specific contexts of the examined P2P schemes. This investigation delineates notable advancements within the P2P energy trading domain yet concurrently brings to light significant lacunae in extant models. A primary concern identified is the absence of standardisation in market mechanisms, which creates inconsistencies across various platforms. That is to say, equally pertinent is the disparity in regulatory maturity observed across different geographical jurisdictions. These issues not only curtail the scalability of P2P platforms but also raise pertinent questions about equity and access within these emerging energy markets.

Consequently, future research should endeavour to expand this scope, exploring operational processes across a wider array of contexts. This could include delving into platforms that solely utilise blockchain technology, thereby enriching our understanding with more comprehensive and diverse insights. Furthermore, the intricacies of regulatory frameworks, crucial for the efficacy and legitimacy of P2P trading, merit detailed examination. Subsequent studies could explore the nuances of regulation within P2P energy trading, drawing valuable comparisons across varied regulatory landscapes.

Additionally, the broader economic context, particularly concerning blockchain and digital currencies such as energy coins, suggests an intriguing research trajectory. Investigating how these economic factors intertwine with existing trading models and innovative approaches like energy credit-based trading schemes could reveal how emerging economic trends are poised to reshape P2P energy trading dynamics. Foremost among these is the application of blockchain and smart contracts, set to revolutionise transactional transparency, security, and efficiency. Concurrently, the advent of artificial intelligence and machine learning is anticipated to significantly enhance energy distribution and dynamic pricing strategies. These technological advancements, coupled with an exploration of regulatory frameworks and policy development across diverse legislative environments, are expected to yield pivotal insights into fostering sustainable P2P energy models. Additionally, the techno-economic analysis of these models will play a critical role in determining their viability and shaping policy decisions.

The integration of sophisticated AI-driven control mechanisms and adaptive learning algorithms is set to elevate the efficiency, security, and economic viability of P2P systems [105]. Innovations such as event-triggered distributed hybrid control schemes suggest a shift towards more responsive and dynamic energy systems [106]. These systems, capable of adjusting energy outputs and parameters effectively in real-time, promise to optimise both security performance and economic operations. Furthermore, the application of hybrid policy-based reinforcement learning in energy-transmission-constrained environments, such as island groups, indicates a future where P2P models are more adaptive

and resilient. This research direction, focusing on AI's transformative potential in energy management, particularly in renewable energy-rich and resource-limited settings, signifies a move towards model-free, adaptive management systems. These advancements, together with an in-depth understanding of consumer behavior, sustainability impacts, scalability, and interoperability considerations, as well as the role of energy storage in maintaining grid stability, will crucially inform the development of future P2P energy trading platforms.

**Author Contributions:** Writing—original draft preparation, S.S.; methodology, S.S. and H.L.; formal analysis, S.S.; writing—review and editing, S.S., H.L., S.Y., V.B. and J.D.; supervision, H.L. All authors have read and agreed to the published version of the manuscript.

**Funding:** This research received no external funding.

**Institutional Review Board Statement:** Not applicable.

**Informed Consent Statement:** Not applicable.

**Data Availability Statement:** The authors confirm that the data supporting the findings of this study are available within the article.

**Conflicts of Interest:** The authors declare no conflict of interest.

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
