# Peer review of "A Case Study of Existing Peer-to-Peer Energy Trading Platforms: Calling for Integrated Platform Features"

_sustainability, doi:10.3390/su152316284_

Round 1

Reviewer 1 Report

Comments and Suggestions for Authors

The paper under review, titled "A case study of existing peer-to-peer energy trading platforms: Calling for integrated platform features," offers a comprehensive review of Peer-to-Peer (P2P) energy trading platforms. It aims to identify key themes that contribute to the functionality and effectiveness of these platforms. While the paper is well-structured and engages with a range of relevant literature, there are several areas where it could be improved to provide a more substantial contribution to the field. The following are specific recommendations aimed at enhancing the depth and rigor of the paper.

  1. Clarification of Objectives: The paper's aim is mentioned but not clearly defined. An elaboration on the specific research questions or hypotheses that guide this review would provide a clearer focus for the reader (Page 1).
  2. Empirical Data: The paper identifies five key themes for P2P energy trading platforms but lacks empirical data to substantiate these themes. Incorporating case studies or empirical data would add depth and practical relevance to your findings (Page 1).
  3. Methodological Details: The methodology section could be more explicit. Providing criteria for the inclusion and exclusion of platforms, as well as the rationale behind focusing on the five identified themes, would enhance the paper's academic rigor (Page 3).
  4. Comparative Analysis: A comparative analysis of the platforms based on the identified themes would offer actionable insights for both researchers and practitioners. This could be integrated into the discussion section to strengthen the paper's contributions (Page 2).
  5. Conclusions: The conclusion section could be more robust by summarizing the key findings succinctly and outlining their implications for the field. Currently, the conclusions serve more as a recap than a synthesis of the paper's contributions (Page 5).

Reviewer 2 Report

Comments and Suggestions for Authors

 This research paper is analysing and comparing seven peer to peer energy trading schemes. It is looking at their mechanisms and five key components ( set-up, market, information, price and regulation).

Comments and suggestions:

Line 285: "...white paper...". It is more common to write the two words with a capital letters.

This paper offers an in-depth analysis of P2P renewable energy trading, which is of a great importance to be explored and expanded over the world.

The context of the whole paper is well organised and presents useful information as to how the main P2P energy trading platforms work, theis similarities and differences. 

However, the energy P2P trading has many limitations and there is a need to further research.

Reviewer 3 Report

Comments and Suggestions for Authors

The paper is clear and easy to follow; however, the authors need to address a couple of details:

1) Table 2 is key to wrap up the study. Despite this, it is not sufficiently discussed and is simply summarized as "Table 2 therefore summarises...". It is highly recommended to add a more complete analysis of this table, especially because the main objective of the paper considers the analysis of case studies as a cornerstone.

2) There are some references that are too old (2013, 2009, 2005, 2004, 2003, 2000, and more, such as 1997, 1994, 1991, 1990, more than 30 years old). With the exception of some seminal author ("father" of some discipline), whose citations can be considered perennial, it is generally recommended that citations be no more than 5 years old.

3) Finally, the authors address 7 case studies, located in different parts of the world; however, from the point of view of the bibliographic review, there are cases located in other places, which although they are not exactly peer-to-peer energy trading platforms, should perhaps have been discussed, such as a couple of examples located in Latin America (e.g. Brazil and Chile):

- Da Costa, R. P., Almeida, C. F. M., Udaeta, M. E. M., Municio, A. L., Nascimento, V. T., & Laurindo, F. J. B. (2022, October). Energy Commercialization Proposition into a P2P Framework in Micro-grids for Brazilian Energy Communities. In 2022 IEEE International Conference on Power Systems and Electrical Technology (PSET) (pp. 280-289). IEEE.

- Condon, F., Franco, P., Martínez, J. M., Eltamaly, A. M., Kim, Y. C., & Ahmed, M. A. (2023). EnergyAuction: IoT-Blockchain Architecture for Local Peer-to-Peer Energy Trading in a Microgrid. Sustainability, 15(17), 13203.

Reviewer 4 Report

Comments and Suggestions for Authors

The manuscript sustainability-2687388 is devoted to the actual scientific problem, namely study of existing peer-to-peer energy trading platforms. The reviewed article is interesting for scholars and theme of the article meets the scope of the journal. Work is performed at sufficient scientific level and has good quality. The manuscript may be considered for publication after minor revision in Sustainability. Prior publication of this manuscript following points needs to be addressed:

  • The quality of Figures 2-5 is not high, it is necessary to improve them.
  • The topic discussed by the authors is quite debatable and can have many directions for development. Therefore, the approach proposed by the authors may have some limitations. To avoid this problem, I propose to add separate section "Limitations and prospects for further study".
  • References list should be carefully checked and journal style policy should be strictly followed (all authors, journal abbreviations, doi, etc).

My decision is minor revision

Comments on the Quality of English Language

Minor editing of English language required

Reviewer 5 Report

Comments and Suggestions for Authors

This paper examines several existing P2P energy trading schemes and analyzes them comparatively.

However, some descriptions are not clear. Some revisions are necessary in the manuscript.

1. Please further elaborate the main innovations and significance of the article.

2. How to choose the right P2P energy trading solution in real engineering applications?

3. Please further explain future research trends in peer-to-peer (P2P) energy trading models.

4. Please further elaborate on the data sources and parameters of the article.

5. In the paper, authors have focused Focus on AI-based methods that can be used in micronetworks. Please compare and analyze different AI methods to improve your work, which can refer to:

[a] IEEE Transactions on Industrial Informatics, vol. 18, no. 2, pp. 835-846, 2022

Comments on the Quality of English Language

A proof reading is needed.

Round 2

Reviewer 1 Report

Comments and Suggestions for Authors

After reviewing the revised manuscript and your responses to previously raised issues, it's commendable to see the efforts you have made to address the feedback. The enhancements in the manuscript have significantly improved its overall quality. Here are some specific comments and suggestions:

1. Clarification of Objectives:

The revision in stating the research questions and hypotheses is well executed. The paper now has a clearer focus, which enhances its academic contribution. The way you have articulated the primary aim and specific research questions is particularly effective.

2. Incorporation of Empirical Data:

The addition of case studies to validate and exemplify the theoretical constructs is a substantial improvement. These practical examples not only add depth but also provide a richer, context-specific understanding of P2P energy trading platforms.

3. Methodological Details:

The expanded methodological section, detailing the inclusion and exclusion criteria for platforms, adds rigor to the study. Your approach to selecting platforms that have been actively engaged in P2P energy transactions and have undergone regulatory scrutiny or peer-reviewed validation is commendable.

4. Comparative Analysis:

Conducting a thorough comparative analysis and integrating it into the discussion section significantly enriches the paper. This comparison offers actionable insights and contributes to a deeper understanding of the topic.

5. Revised Conclusions:

The revised conclusion is a bit lengthy. Try to distill the conclusion to the most critical findings and implications. This will allow readers to grasp the essence of your research more quickly.

Comments on the Quality of English Language

The quality of English in the manuscript is generally good, demonstrating a high level of proficiency. The language is clear, the terminology is appropriate for the field, and the sentence structure is well-composed, facilitating ease of understanding.

Reviewer 5 Report

Comments and Suggestions for Authors

No further comments. 

Author Response

Thank you for your feedback!